# Single Image Dehazing Using Global Illumination Compensation

**DOI:** 10.3390/s22114169

**Published:** 2022-05-30

**Authors:** Junbao Zheng, Chenke Xu, Wei Zhang, Xu Yang

**Affiliations:** School of Information Science and Technology, Zhejiang Sci-Tech University, Hangzhou 310018, China; zhengjunbao@zstu.edu.cn (J.Z.); xuchenke98@163.com (C.X.); yangxu@zstu.edu.cn (X.Y.)

**Keywords:** fog imaging, image dehazing, shading partition enhancement mechanism, global illumination compensation factor, dark channel prior (DCP), guided filtering

## Abstract

The existing dehazing algorithms hardly consider background interference in the process of estimating the atmospheric illumination value and transmittance, resulting in an unsatisfactory dehazing effect. In order to solve the problem, this paper proposes a novel global illumination compensation-based image-dehazing algorithm (GIC). The GIC method compensates for the intensity of light scattered when light passes through atmospheric particles such as fog. Firstly, the illumination compensation was accomplished in the CIELab color space using the shading partition enhancement mechanism. Secondly, the atmospheric illumination values and transmittance parameters of these enhanced images were computed to improve the performance of atmospheric-scattering models, in order to reduce the interference of background signals. Eventually, the dehazing result maps with reduced background interference were obtained with the computed atmospheric-scattering model. The dehazing experiments were carried out on the public data set, and the dehazing results of the foggy image were compared with cutting-edge dehazing algorithms. The experimental results illustrate that the proposed GIC algorithm shows enhanced consistency with the real-imaging situation in estimating atmospheric illumination and transmittance. Compared with established image-dehazing methods, the peak signal-to-noise ratio (PSNR) and the structural similarity (SSIM) metrics of the proposed GIC method increased by 3.25 and 0.084, respectively.

## 1. Introduction

Haze is an atmospheric phenomenon where dust, smoke and other dry particles obscure the clarity of the atmosphere. This results in a loss of contrast, visibility and vividness in images required for vision technology [1,2] due to the scattering effect of light through haze particles. Therefore, in applications involving image sensors and related image processing, such as video surveillance, assisted driving, etc., it is important to dehaze images to improve target recognition capabilities particularly.

The single image-dehazing technology in recent years can be divided into three categories: (1) The image-dehazing method based on a physical model. Under the framework of the atmospheric-scattering model [3], prior knowledge has been employed to estimate the transmittance and atmospheric light intensity and then combined with the model to dehaze the image. By observing a large number of foggy images, He et al. [4] found that the intensity of some pixels in at least one-color channel of the foggy image is very low, close to zero, and proposed a fast image-dehazing method based on dark channel prior (DCP). DCP provides an effective method to estimate model parameters, but it still suffers from slow processing speed and unpredictable performance in the sky regions. Since then, there have been many improvements to DCP. Dhara et al. [5] used weighted least-squares filtering and color correction on DCP to optimize the dehazing effect. Kim [6] proposed a sky detection method using region-based and boundary-based sky segmentation, which enables DCP to perform image restoration for sky of various shapes. Chung et al. [7] estimated, by analyzing a large number of optimal scattering coefficients and the distribution between dark channels, an appropriate scattering coefficient. Simultaneously, dehazing algorithms based on new prior knowledge [8,9,10,11] have emerged. (2) The image-dehazing method based on image fusion. Ancuti et al. [12] proposed an image-enhancement method based on image fusion that excludes the atmospheric-scattering model. This method obtains and filters the information in the original image through various methods, and then fuses Laplacian and Gaussian pyramids to achieve the effect of dehazing. On this basis, Choi et al. [13] completed dehazing by extracting haze-related features and using a weighting scheme that selectively fuses images. Cho et al. [14] introduced the adaptive tone remapping (ATR) algorithm to achieve balanced image enhancement while refining image details. Ngo et al. [15] took a set of detail-enhanced and under-exposed im-ages derived from a single haze image as the input for image fusion, and combined the ATR algorithm to complete dehazing. (3) The image-dehazing method based on deep learning. The success of deep learning in computer vision tasks has led to a large number of deep-learning-based dehazing methods, such as convolutional neural networks (CNN) [16], generative adversarial networks [17,18,19], attention-based multi-scale models [20,21], and encoder–decoder structured networks [22].

Among the above methods, although the fusion-based method guarantees the gradient information of multi-scale input, its image restoration strength is insufficient to some degree. The deep-learning-based methods usually need considerable well-labeled samples, indicating high cost in data collection and curation. Based on the atmospheric-scattering model, we analyze the shortcomings of the DCP method and the fog-imaging principle. It is believed that the brightness and color of a pixel of the input image will be affected by the surrounding pixels in a limited range. Therefore, in order to eliminate or reduce the interference from the background and obtain the real prior information, this paper proposes the GIC algorithm. Before estimating atmospheric illumination and transmittance, the GIC algorithm eliminates or weakens the interference of the background on the color distribution of the image, thereby obtaining more accurate model parameters. Our main contributions are summarized as follows:

(1) We propose an image-dehazing method based on global illumination compensation. Compared with other image-dehazing methods, the GIC method eliminates or reduces the influence of surrounding pixels on the target pixel through the shading partition enhancement mechanism before performing a priori analysis. In evaluation experiments on public datasets, the GIC method achieves better image evaluation metrics than the candidate methods, yet keeps high computational efficiency.

(2) In order to eliminate the interference from the background in the process of dehazing, the GIC algorithm employs a contrast-enhancement technique named the shading partition enhancement mechanism to distinguish the light and dark information of the image. The shading partition enhancement mechanism calculates the real atmospheric illumination observation points and eliminates or reduces the interference of surrounding pixels to target pixels by means of local enhancement.

(3) The GIC algorithm uses the global illumination compensation factor and guided filtering for transmittance map refinement to ensure a dehazed image with no halo artifacts. The global illumination compensation factor is a local weighted average algorithm that can make the transmission map have sharper edges.

The remainder of this paper is organized as follows: In Section 2, the atmospheric-scattering model, dark channel prior, as well as the guided filtering method are introduced as background information about image dehazing. In Section 3, the GIS approach is proposed, in detail, to overcome the drawbacks of established dehazing methods. Section 4 contains its subjective and quantitative comparison with the state-of-the-art dehazing methods, before a conclusion is drawn in Section 5.

## 2. Problem Formulation of Image Dehazing

In this section, we introduce the atmospheric-scattering model which is the basic underlying model of image dehazing. Then we introduce some existing methods used in this paper to calculate haze-related parameters. Finally, we analyze the foggy imaging process that inspired the idea of this paper.

### 2.1. Atmospheric-Scattering Model

The atmospheric-scattering model [3] describes the dehazing process of haze images and is widely used in the field of image dehazing. According to the atmospheric-scattering model, the formation of blurred images can be described by Equation (1):(1)I(x)=J(x)t(x)+(1−t(x))A
where I(x) is the observed intensity, J(x) is the scene radiance, A is the global atmospheric light, and t(x) is the medium transmission describing the portion of the light that is not scattered and reaches the camera. It can be expressed as $t(x)=e^{-\beta d(x)}$, where β is the attenuation coefficient and t(x) indicates the scene depth. The real scene can be recovered after A and t(x) are estimated.

### 2.2. Calculation of Haze-Related Parameters

This subsection mainly introduces the two methods used in this paper to calculate the haze-related parameters.

#### 2.2.1. Dark Channel Prior

The dark channel prior method is mainly designed for outdoor haze-free images. In these non-sky areas, there exists one-color channels that have pixels with intensity close to zero. The dark channel is defined as the minimum value of the RGB of pixels in the local area of the image:(2)$$J^{dark}=\underset{y\in \Omega (x)}{\min}(\underset{c\in \{r,g,b\}}{\min}J^{c}(y))$$
where $J^{c}$ is a color channel of J, c∈{r,g,b} is any one of the three channels of the image, and Ω(x) is a local patch centered at x. The value of the dark channel is related to the fog density of the image and can be used to closely estimate atmospheric illumination and transmittance.

#### 2.2.2. Guided Filtering Method

Guided filtering [23] is a novel explicit image filter. Derived from a local linear model, the guided filter computes the filtering output by considering the content of a guidance image. The key assumption of the guided filter is a local linear model between the guidance I and the filtering output q. This method assumes that q is a linear transform of I in a window $\omega_{k}$ centered at the pixel k:(3)$$q_{i}=a_{k}I_{i}+b_{k},\forall i\in \omega_{k}$$
where q is the value of the output pixel, I is the pixel value of the input image, i and k are the pixel indices, $a_{k}$ and $b_{k}$ are the coefficients of the linear function when the center of the window is at k.

To determine the linear coefficients $a_{k}$ and $b_{k}$, we need constraints from the filtering input p, which can be derived from Equations (4) and (5).
(4)$$a_{k}=\frac{\frac{1}{\left|\omega \right|}\sum {}_{i\in \omega_{k}}I_{i}p_{i}-\mu_{k}{\bar{p}}_{k}}{2}$$
(5)$$b_{k}={\bar{p}}_{k}-a_{k}\mu_{k}$$
where $\mu_{k}$ and $\sigma_{k}^{2}$ are the mean and variance of I in $\omega_{k}$, |ω| is the number of pixels in $\omega_{k}$, ${\bar{p}}_{k}=\frac{1}{\left|\omega \right|}\sum {}_{i\in \omega_{k}}p_{i}$ is the mean of p in $\omega_{k}$. Having obtained the linear coefficients $a_{k}$ and $b_{k}$, we can compute the filtering output $q_{i}$ by Equation (3). This model can be used to optimize the initial transmittance.

### 2.3. Imaging Process under Haze Conditions

Under various haze imaging conditions, the image captured by the camera is formed by the light reflected from the object penetrating the air diffusely. There are three main sources of light hitting an object. The first source indicates that the light is irradiated on the object by the light source directly. In the second situation, the light is reflected on the object after the light source irradiates the surrounding object diffusely. In the third case, the diffuse reflection light of the surrounding objects is irradiated on the atmospheric particles and then scattered to the object. Shown in Figure 1, the picture captured by the camera is formed by the direct lighting and the indirect lighting reflected by objects and atmospheric particles.

Model-based methods often estimate atmospheric illumination and transmittance by analyzing the local and global color distribution of the input foggy image. However, it can be seen from Figure 1 that a certain pixel of the real image is affected by atmospheric particles and surrounding pixels. The traditional model-based methods directly perform prior analysis on the input image (disturbed image), and the following problems arise: (1) The estimated atmospheric light value A is small due to the reflection that occurs when light from the light source passes through atmospheric particles such as fog. (2) Transmittance t(x) refers to the part of the light reflected by the target pixel that is not scattered during the process of projecting the light from the target pixel to the camera through the medium. The influence of the surrounding pixels on the target pixel causes the part received by the camera to change, resulting in an inaccurate estimated t(x). Therefore, it is necessary to eliminate or reduce the interference from the background to estimate the correct atmospheric illumination and transmittance.

## 3. The Proposed GIS Method

The atmospheric-scattering model in Section 2.1 suggests that estimation of the atmospheric illumination and transmittance are two crucial steps to recover a haze-free image. In order to obtain more accurate atmospheric illumination values and transmittance, we propose GIC, an image-dehazing algorithm based on atmospheric-scattering model and considering image background interference. A few intermediate outputs in our procedure to obtain J from I using GIC method are shown in Figure 2. The GIC algorithm contains two key steps, i.e., the shading partition enhancement and global illumination compensation factor. In this section, we focus on the method design of these two steps and the evaluation indexes used in this paper. The detailed steps of the GIC algorithm are shown in Algorithm 1.
**Algorithm 1.** Global Illumination Compensation-Based Image-Dehazing algorithm.**Input:** foggy image I**.****Procedure:**I is processed using the shading partition enhancement mechanism.   **#Mechanism 1** The Shading Partition Enhancement Mechanism   Get the L-channel image matrix L of I in the CIELab color space and normalize it.   Obtain the local bright and dark matrices Lw, Lb by Equation (6).   Estimate the range of bright and dark areas k1 and k2 by Equations (7) and (8).   Obtain the bright and dark area information b and d through k1 and k2 by Equations (7) and (8).   Combine b, d, and intermediate region information to get enhance L* by Equation (9).   Convert back to RGB color space to get enhanced image I*.Perform DCP calculations on I* to estimate atmospheric illumination A and initial transmittance t(x).Calculate the global illumination compensation factor ω(x).   **#Mechanism 2** Global Illumination Compensation Factor   Perform DCP calculations on I*.   Calculate the mean over a small range centered at pixel x as a factor at x and write it to ω(x).Optimize t(x) with guided filtering and ω(x).Combine A and t(x) to dehaze by Equation (13).**Output:** clear image J.

### 3.1. The Shading Partition Enhancement

He et al. [4] first obtained the pixel position with the largest gray value of 0.1% from the dark channel image. Then, they found the pixels at these locations in the input image and used the maximum brightness value of these pixels as the atmospheric illumination value. We calculated these pixel positions in the synthetic objective testing set (STOS) [24] dataset by this method. We found that the brightness standard deviation of surrounding pixels tended to fluctuate within a small range. After adding the brightness value of the center pixel, the standard deviation increased significantly. Combined with the experimental results, we believe that the sudden increase in the brightness value of the center pixel was affected by the surrounding pixels. In order to find the correct measurement point of atmospheric illumination, this paper proposes the shading partition enhancement mechanism [25].

The shading partition enhancement mechanism processes the light and dark areas of the L-channel of the input image, in order to enhance the contrast of the image in the light and dark areas. The basic process of shading partition enhancement is shown in Figure 3.

In the first step, the enhancement mechanism normalizes the L-channel of the input image, and obtains the largest (or smallest) pixel from the L-channel using an n × n window (here n = 3). The window slides by 1 pixel each time. The maximum value (or minimum value) of each window area is used to obtain a local bright (dark) matrix. Subsequently, an n × n window (here n = 3) was used to slide on the local light (dark) matrix. For each position, the product of other elements around the central element is calculated to form a single output matrix element. The calculation process is shown in Equation (6). Based on computation, the brighter areas in the input image are enhanced and displayed in Lw. As the enhanced dark areas are barely visible, this part will be inverted to show in Lb.
(6)$$\left\{ \begin{array}{l} Lw=\prod_{y\in \Omega (x)}\underset{y\in \Omega (x)}{\max}(I_{L}(y))\\ Lb=-\prod_{y\in \Omega (x)}\underset{y\in \Omega (x)}{\min}(I_{L}(y)) \end{array}\right.$$

The next step is the process of fusing the enhanced light and dark area information with the input image. The light and dark area information needs to be obtained from Lw and Lb. As shown in Equations (7) and (8), where bright and dark are the light and dark area information, $k_{1}$ and $k_{2}$ are the light and dark area ranges of the L-channel, respectively.
(7)$$\left\{ \begin{array}{l} k_{1}=\max(L)-mean(L)\\ bright=\frac{k_{1}(Lw-mean(Lw))}{\max(Lw-mean(Lw))} \end{array}\right.$$
(8)$$\left\{ \begin{array}{l} k_{1}=\min(L)-mean(L)\\ dark=\min(\frac{-k_{2}Lb}{\max(Lb)})-\frac{-k_{2}Lb}{\max(Lb)} \end{array}\right.$$

At this time, the enhanced light and dark area information has been obtained. The previous image preprocessing process removed the intermediate area information. In the later fusion, the intermediate area information needs to be filled back. The fused image information is shown in Equation (9). Finally, we go back from the Lab color space to the RGB color space to get our enhanced image. This mechanism can eliminate or reduce the influence of surrounding pixels on the target pixel effectively.
(9)$$L^{*}=\left\{ \begin{array}{l} bright+mean(L),bright\geq 0\\ dark+mean(L),bright<0 \end{array}\right.$$

After eliminating the influence of surrounding pixels, the atmospheric illumination value is calculated by [4]. The experimental results show that the atmospheric illumination value estimated by the enhanced image is more in line with the real imaging situation.

### 3.2. Global Illumination Compensation Factor

The GIC method first calculates DCP on the enhanced image to obtain the atmospheric illumination value, then performs the optimization of the transmittance.

With the computed atmospheric illumination values, the DCP principle is used to calculate the dark channel of the atmospheric-scattering model:(10)$$\underset{y\in \Omega (x)}{\min}(\underset{c\in \{r,g,b\}}{\min}\frac{I^{c}(y)}{A^{c}})=t(x)\underset{y\in \Omega (x)}{\min}(\underset{c\in \{r,g,b\}}{\min}\frac{J^{c}(y)}{A^{c}})+1-t(x)$$

According to the DCP principle, the dark channel of the image should be close to zero under fog-free conditions, so there are:(11)$$t(x)=1-\underset{y\in \Omega (x)}{\min}(\underset{c}{\min}\frac{I^{c}(y)}{A^{c}})$$

After eliminating or reducing the influence from the surrounding scene through the shading partition enhancement mechanism. The overall outline of the calculated initial transmittance map is not clear enough. Therefore, we introduce a global illumination compensation factor on the basis of the enhanced dark channel image, with the purpose of optimizing the transmittance. Assuming that the target pixel is affected by the surrounding pixels in a local area and the average information of the surrounding pixels is consistent, this paper introduces a weight coefficient to t(x), namely:(12)$$t(x)=\omega (x)\left[1-\underset{y\in \Omega (x)}{\min}(\underset{c}{\min}\frac{I^{c}(y)}{A^{c}})\right]$$
where $\omega (x)=\underset{y\in \Omega (x)}{mean}(\underset{y\in \Omega (x)}{\min}(\underset{c}{\min}I^{{*}^{c}}))$, refers to the average information matrix of pixels in the local area centered on x in the enhanced dark channel image. This can enhance, to a certain extent, the profile characteristics of the transmittance map.

### 3.3. Dehazing Process of the GIC Algorithm

In the GIC algorithm, we obtain the enhanced image through the shading partition enhancement mechanism and combine the DCP to obtain the atmospheric illumination value from the enhanced image. To optimize the initial transmittance, we use global illumination compensation factor and guided filtering to smooth the image and highlight image edges. By Equation (13), the haze-free image can be recovered easily.
(13)$$J(x)=\frac{I(x)-A}{t(x)}+A$$

### 3.4. Evaluation Indexes of Image Dehazing

To evaluate the dehazing performance quantitatively, two popular metrics are adopted in this study: PSNR and SSIM [26]. Given a ground-truth image Y and a processed image X, definition PSNR is defined given in Equation (14), where *MSE* (mean square error) is the mean square error of the processed image X and the ground-truth image Y; n is the number of bits per pixel.
(14)$$PSNR=10\log_{10}(\frac{{\left(2^{n}-1\right)}^{2}}{MSE})$$

The SSIM index measures image similarity in terms of brightness, contrast, and structure. Its calculation method is shown in Equation (15) where $\mu_{x}$ and $\mu_{y}$ are the mean value of image X and Y, $\sigma_{x}$ and $\sigma_{y}$ are the standard deviation of image X and Y, and $\sigma_{xy}$ is the covariance of image X and Y.
(15)$$SSIM(x,y)=\frac{\left(2\mu_{x}\mu_{y}+c_{1})(\sigma_{xy}+c_{2}\right)}{\left(\mu_{x}^{2}+\mu_{y}^{2}+c_{1})(\sigma_{x}^{2}+\sigma_{y}^{2}+c_{2}\right)}$$

However, a completely clear haze-free image cannot be obtained in reality and the two indexes themselves are limited and difficult to keep consistent with the quality of human perception. Therefore, two no-reference IQA indexes were selected—spatial and spectral entropies quality (SSEQ) [27] and natural image quality evaluator (NIQE) [28] to supplement the shortage of PSNR and SSIM. SSEQ calculates the quality of the distorted image through the image spectral probability map, and its calculation method is shown in Equation (16), where *P*(*i*,*j*) is the probability map of the pixel in the spectrum.
(16)$$E_{f}=-\sum_{i}\sum_{j}P(i,j)\log_{2}P(i,j)$$

NIQE represents the quality of the distorted image by the distance between the natural scene statistic (NSS) feature model and the multivariate Gaussian (MVG) extracted from the distorted image features. The calculation method of this indicator is shown in Equation (17), where $\nu_{1}$, $\nu_{2}$, $\sum_{1}$, $\sum_{2}$ are the mean vector and covariance matrix of MVG.
(17)$$D(\nu_{1},\nu_{2},\sum_{1},\sum_{2})=\sqrt{{\left(\nu_{1}-\nu_{2}\right)}^{T}{\left(\frac{\sum_{1}+\sum_{2}}{2}\right)}^{-1}(\nu_{1}-\nu_{2})}$$

Note that a higher PSNR and SSIM value and a lower SSEQ and NIQE value indicate higher quality in the dehazing process.

## 4. Experimental Outcomes and Discussion

In order to verify the effectiveness of the shading partition enhancement mechanism in image dehazing, we analyzed its correctness for obtaining atmospheric illumination observation points and compared it with the established dehazing methods, including Dhara [5], Berman [9], Choi [13], Cho [14], and Cai [16].

### 4.1. Analysis of the Shading Partition Enhancement Mechanism

In the STOS dataset, the position of the observation point of the atmospheric illumination value and the brightness value of the surrounding pixels were calculated in [4] and the GIC algorithm. In more than 70% of the images, the average brightness of surrounding pixels calculated by the GIC algorithm was higher than in the literature [4]. In 74.6% of the images, the average brightness of the observation points and surrounding pixels of the GIC algorithm only fluctuated around 0.3, showing better stability than the literature [4]. The brightness reflects the fog concentration around the observation point. It can be seen that the observation points of atmospheric illumination value calculated by the GIC algorithm was more in line with the real imaging situation.

In order to verify the effectiveness of the light–dark partition enhancement mechanism proposed in this paper in image dehazing, we compared it with various image enhancement algorithms. Method B1 does not perform image enhancement, method B2 uses histogram equalization processing, method B3 denotes contrast-limited adaptive histogram equalization (CLAHE) [29] processing, and method B4 adapts Retinex [30] processing. Except for the different image-enhancement methods, the other processes of these methods are the same as the GIC algorithm. Figure 4 shows the dehazing effect of an outdoor real foggy image in the SOTS dataset. After GIC algorithm processing, the color of the trees in the first picture was greener, the outline was clearer, and the scenery in the second picture was clearer. Table 1 shows the average value (AVG) and standard deviation (SD) of the reference indexes obtained after dehazing all the fog images in the SOTS dataset by the five methods. The average PSNE and SSEQ indexes of the GIC method were significantly higher than other methods. Standard deviations of SSIM and NIQE indexes obtained by the GIC method during dehazing were the lowest among selected approaches, demonstrating stable performance. From both qualitative and quantitative perspectives, the shading partition enhancement mechanism had better dehazing effect and stability than other image-enhancement methods, proving the effectiveness of this mechanism in image dehazing. Due to limited impact on image quality, shading partition enhancement can be regarded as a suitable preprocessing mechanism for this SOTS dataset, which contains considerable outdoor images.

### 4.2. Dehazing of Indoor Simulated Images

Indoor synthetic foggy images were used in the SOTS dataset to conduct simulation experiments to evaluate the feasibility of the above algorithms from both subjective and objective aspects. Image-dehazing algorithms of Dhara [5], Berman [9], Choi [13], Cho [14], and Cai [16] were selected for comparison. The results of these algorithms are produced by the author’s code with the specified parameters. Dehazing results with those dehazing methods of indoor images are illustrated in Figure 5.

It can be observed from Figure 5 that for indoor images at different concentrations, Dhara [5], Choi [13], and Cho [14] algorithms achieved good results in dehazing. However, the dehazing effect of these algorithms came at the expense of image color fidelity. The GIC method, Berman [9], and Cai [16] all had better processing effects on haze images. In the red area in the upper left corner of the image, the GIC method had more saturated colors compared to Berman [9] and Cai [16]. In the processing of indoor synthetic simulation images, the GIC method had better color retention ability for colored areas and brightly colored mist images. After obtaining better results subjectively, we used SSEQ to evaluate the quality of the image after dehazing.

Considering that the fog concentration has a direct impact on the image-dehazing effect, ten different concentrations were set in the verification process to test the algorithm performance. For the same indoor images under ten different fog concentrations, we used the above methods to dehaze the sample image set respectively, as shown in Figure 6.

In general, high levels of fog tend to a negative impact on the quality of image dehazing. With the increase of fog concentration, except for the SSEQ index obtained by the Cho [14] method which increased slightly, the SSEQ index obtained by the other five dehazing methods showed a downward trend generally. The SSEQ index obtained by the GIC method was the lowest among the selected methods and maintained a continuous downward trend. The GIC method can better ensure the quality of the image, and there is no obvious loss of image quality due to the enhancement of image contrast, which proves the feasibility of the GIC method.

### 4.3. Dehazing of Outdoor Real Images

After processing the simulated images to achieve good results, we conducted experiments on real outdoor foggy images in the SOTS dataset, and compared the results with the dehazing algorithms of Dhara [5], Berman [9], Choi [13], Cho [14], and Cai [16]. The effectiveness of the dehazing algorithm in this paper was evaluated from the image quality evaluation index.

In the visual analysis, we increased the dehazing effect of the He method. The comparison results of outdoor images are shown in Figure 7. As can be seen from Figure 7, after the six candidate methods and the GIC algorithm processed the input image, the clarity and contrast of the image were improved to different degrees. He [4] suffered from halo artifacts where the sky meets the object. Although Dhara [5], Choi [13] and Cho [14] had higher dehazing intensity, they all showed striped color distortion. The dehazing effect of Berman [9] and Cai [16] was weaker among the methods. Compared with the six candidate methods, the GIC algorithm removed the fog in the image effectively. There is no image distortion in the dehazing result. However, in general, there was still a shortage of incomplete fog removal.

Table 2 shows the comparison of evaluation indexes of outdoor image-dehazing processing. From the PSNR and SSIM indexes reflecting the dehazing effect, the GIC method outperformed the other five candidate methods. In terms of NIQE index, the Cai [16] method was better slightly; Cho [14] achieved the lowest SSEQ index, but the NIQE index obtained by this method was significantly higher than that of other candidate methods, indicating that this method is not balanced in performance. According to the results in Table 1, it can be seen that the GIC method not only had a better dehazing effect, but also had a stable performance relatively, maintaining a good balance in different performance indexes. The above results show that the GIC method has better restoration effect.

The GIC method preprocesses the input image through the shading partition enhancement mechanism, which compensates for the light intensity scattered by atmospheric particles such as fog. Meanwhile, the influence of surrounding pixels on the target pixel is also eliminated. Realistic observation points of atmospheric illumination can be found in the enhanced image. The global illumination compensation factor optimizes transmittance in order to obtain a sharper profile transmittance map. In this way, the GIC method completes the fog removement with the atmospheric-scattering model. Experimental results show that the GIC method had improved performance on outdoor foggy images, illustrated by reduced image distortion and darkening in the restoration result.

We also test the computational efficiency of candidate dehazing methods, as shown in Table 3, on the computer with i5-1135G7 CPU and 16 GB of running memory, the time required to run all the images in the SOTS dataset. The dehazing method proposed by Berman [9] and Choi [13] had low computational efficiency in the SOTS dataset, and the GIC method had the highest running efficiency, which proves the effectiveness and high performance of the algorithm.

## 5. Conclusions

Under the framework of the atmospheric-scattering model, this paper designs and implements the GIC algorithm. Introducing the global illumination factor compensation and the shading partition enhancement can eliminate or reduce the interference from the background effectively and improve the image-dehazing quality and effect. The experimental results show that the proposed method can effectively reduce the background interference through the shading partition enhancement and the global illumination compensation factor, and the obtained atmospheric illumination and transmittance are more in line with the real situation during imaging. From the perspective of quantitative analysis, the GIC method can achieve higher PNSR and SSIM indexes for outdoor scene images, which are better than the selected candidate algorithms; it has a better dehazing effect on indoor low-density foggy images. From the perspective of subjective observation, the GIC algorithm can maintain the natural color of the image well and produce clear dehazing images with high operating efficiency.

## Figures and Tables

**Figure 1 sensors-22-04169-f001:**
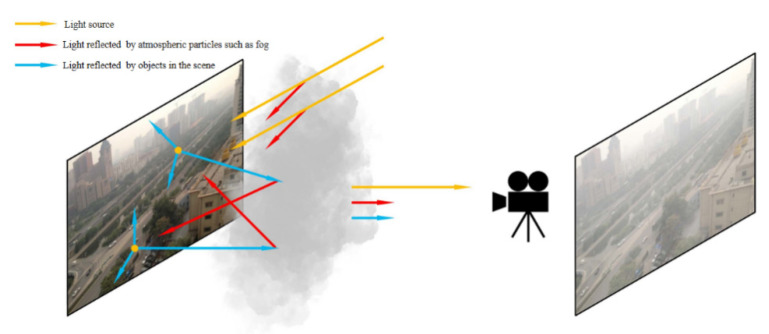
Imaging process under haze conditions.

**Figure 2 sensors-22-04169-f002:**
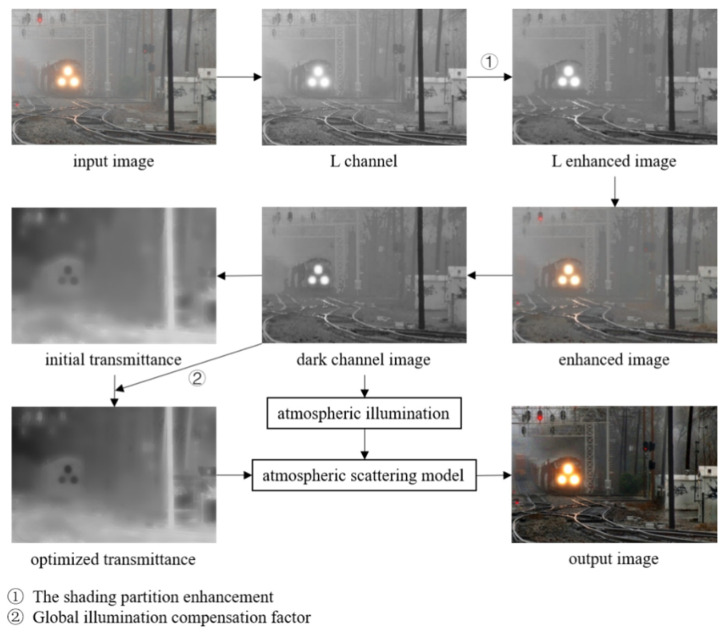
The flow chart of the proposed GIC algorithm. Steps 1 and 2, denoted as the shading partition enhancement and compensation factor, are two key components employed in the GIC algorithm to improve the image-dehazing performance. For dark channel images, step 2 aims to obtain optimized transmittance using a global illumination compensation factor.

**Figure 3 sensors-22-04169-f003:**
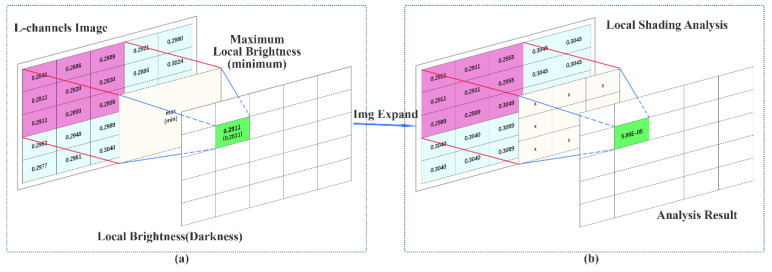
The shading partition enhancement consists of two parts: (**a**) used to obtain the maximum (minimum) pixel value in the local area of the input image and (**b**) the center pixel is enhanced by surrounding pixels.

**Figure 4 sensors-22-04169-f004:**
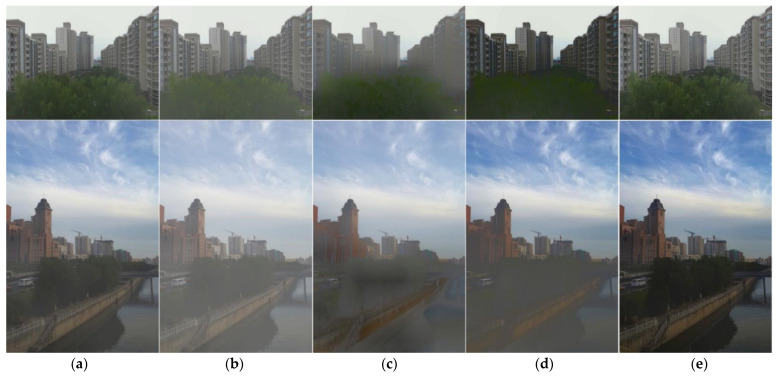
Comparison of outdoor image dehazing for the five methods. (**a**) B1 method processing results, (**b**) B2 method processing results, (**c**) B3 method processing results, (**d**) B4 method processing results, and (**e**) GIC method processing results.

**Figure 5 sensors-22-04169-f005:**
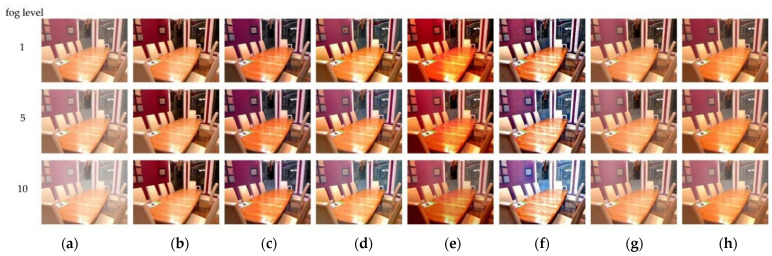
Dehazing effects of various methods on indoor composite images under different fog concentrations. (**a**) Input images, (**b**) fog-free image, (**c**) Dhara method, (**d**) Berman method, (**e**) Choi method, (**f**) Cho method, (**g**) Cai method, and (**h**) the proposed GIC method.

**Figure 6 sensors-22-04169-f006:**
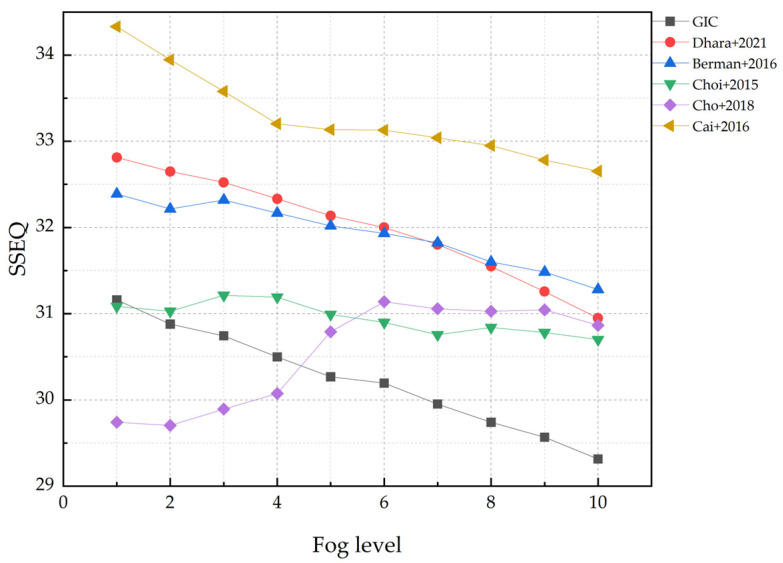
The SSEQ indicator changes of each method at multiple fog levels.

**Figure 7 sensors-22-04169-f007:**
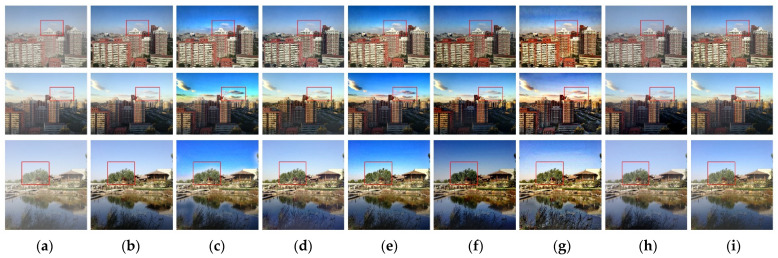
Comparison of the results of different dehazing methods for outdoor real images. (**a**) Input image, (**b**) fog-free image, (**c**) He method processing results, (**d**) Berman method processing results, (**e**) Dhara method processing results, (**f**) Choi method processing results, (**g**) Cho method processing results, (**h**) Cai method processing results, and (**i**) GIC method processing results.

**Table 1 sensors-22-04169-t001:** Comparison of indexes of the five methods.

Methods	PSNR		SSIM		SSEQ		NIQE	
	AVG	SD	AVG	SD	AVG	SD	AVG	SD
B1	18.67	3.61	0.88	0.05	18.20	6.36	2.61	0.60
B2	18.48	3.25	0.79	0.09	18.01	6.30	3.39	0.81
B3	19.54	2.39	0.71	0.09	16.25	5.87	4.14	0.81
B4	13.75	4.10	0.51	0.24	24.65	15.10	4.09	1.72
GIC	**22.07**	**3.02**	**0.90**	**0.05**	**15.49**	**5.60**	2.76	**0.59**

The data in bold in the table represent the best index for each column.

**Table 2 sensors-22-04169-t002:** Comparison of evaluation indexes in image dehazing.

Methods	PSNR		SSIM		SSEQ		NIQE	
	AVG	SD	AVG	SD	AVG	SD	AVG	SD
Dhara [5]	17.13	3.35	0.83	0.08	14.87	5.70	2.75	0.58
Berman [9]	18.33	3.13	0.79	0.09	15.47	6.26	2.80	0.67
Choi [13]	18.97	3.69	0.83	0.07	15.53	6.01	2.80	0.67
Cho [14]	17.78	**2.32**	0.74	0.08	**14.18**	8.27	3.26	1.95
Cai [16]	21.90	3.09	0.89	0.08	17.96	7.38	**2.66**	**0.57**
GIC	**22.07**	3.02	**0.90**	**0.05**	15.49	**5.60**	2.76	0.59

The data in bold in the table represent the best index for each column.

**Table 3 sensors-22-04169-t003:** The efficiency of candidate dehazing methods.

Method	Berman [9]	Dhara [5]	Choi [13]	Cho [14]	GIC
time (s)	2278.5	302.22	5605	307.92	140.44

## Data Availability

Not applicable.

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
