# Peer review of "Single Image Dehazing Using Global Illumination Compensation"

_sensors, 2022, doi:10.3390/s22114169_

Round 1

Reviewer 1 Report

This paper presents a work on developing a single image dehazing method by using global illumination compensation.  There are several suggestions for improvements:

1) All abbreviations should be defined at their first used in the abstract or in text.  For examples, in the abstract, what are GIC, SOTS, PSNR and SSIM?

2) Better to have a paragraph between Section 2 with Section 2.1.

3) Better to have a paragraph between Section 2.2 with Section 2.2.1.

4) Mathematical symbols should be typed in Italic.  For examples, on page 5, L and n should be in Italic.

5) Section 2.1 is lacking of citation.  Please add the corresponding citation.

6) In Section 2.2, from my knowledge, there are more than only two methods for dehazing.  But why Section 2.2 only limits to only two dehazing methods?

7) Section 2.2.1 is actually using the theory from Section 2.1.  How about Section 2.2.2?  How it relates with equation (1)?

8) In Section 2.2.1, it is better if the equation showing how the dehazed image is obtained from equation (2) is provided.

9) In equation (3), seems that coefficients a and b are locally determined, based on k.  How those values are calculated?  Better to give formulas.

10) What are Lw and Lb in equation (4)?

11) In equation (7), what will happen if bright equal to zero?

12) In equation (12), what is lg? Is this log_2 or log_10?

13) In line 246, it is claimed that the method is compared with "state-of-art methods".  Yet, [4] is from year 2016, [7] is from year 2021, [10] is from year 2015, [11] is from year 2018, and [12] is from year 2016.  From this, only ref. [7] is recent, whereas others are old methods.  Better to add more recent methods for comparisons (e.g., methods from year 2021 and 2022).

14) Methods in line 246 and 247 should be introduced properly in Section 1.  It would be nice if the authors could discuss some limitations of these methods in section 1.

15) References are mostly old.  Better to add some more recent references, and discuss them in Section 1.  Some examples:

https://doi.org/10.3390/s20185170

https://doi.org/10.3390/s21237922

 https://doi.org/10.3390/electronics10212636

Author Response

Response to Reviewer 1 Comments

Point 1: All abbreviations should be defined at their first used in the abstract or in text.  For examples, in the abstract, what are GIC, SOTS, PSNR and SSIM?

Response 1: GIC denotes the proposed Global Illumination Compensation based Image Dehazing algorithm (GIC). STOS is a benchmark dataset to validate the effectiveness of dehazing approaches. As PSNR and SSIM indexes are evaluate metrics of image dehazing.

The Abstract has been modified as follows:

“The existing dehazing algorithms hardly consider background interference in the process of estimating the atmospheric illumination value and transmittance, resulting in an unsatisfactory dehazing effect. In order to solve the problem, this paper proposes a novel Global Illumination Compensation based Image Dehazing algorithm (GIC). … Compared with the average metrics of each method, the peak signal-to-noise ratio (PSNR) and the structural similarity (SSIM) metrics have increased 3.25,0.084 respectively.”

Point 2: Better to have a paragraph between Section 2 with Section 2.1.

Response 2:

To describe the content of Section 2 more clearly, add content to Page 3, Lines 108-111.

In Page 3, Lines 108-111.

The following sentences have been added

“In this section, we introduce the atmospheric scattering model which is the basic underlying model of image dehazing. Then we introduce some existing methods used in this paper to calculate haze-related parameters. Finally, we analyze the foggy imaging process that inspired the idea of this paper.”

Point 3: Better to have a paragraph between Section 2.2 with Section 2.2.1.

Response 3:

To describe the content of Section 2.2 more clearly, add content to Page 3, Lines 122-123.

In Page 3, Lines 122-123.

The following sentences have been added

 “This subsection mainly introduces the two methods used in this paper to calculate the haze-related parameters  and .”

Point 4: Mathematical symbols should be typed in Italic.  For examples, on page 5, L and n should be in Italic.

Response 4:

L refers to the L-channel of the CIELab color space, not mathematical notation. "L channel " in the text has been modified as "L-channel ".

In Page 6, Lines 212 and 214.

The n has been modified as  .

Point 5: Section 2.1 is lacking of citation.  Please add the corresponding citation.

Response 5:

In Page 3, Lines 113.

“The atmospheric scattering model” has been modified as “The atmospheric scattering model [3]”.

Point 6: In Section 2.2, from my knowledge, there are more than only two methods for dehazing.  But why Section 2.2 only limits to only two dehazing methods?

Response 6:

I made a typo here; my original intention was to write some algorithmic backgrounds used in the following sections.

In Page 3, Lines 121.

The title of Section 2.2 has been modified as Calculate Haze-related Parameters Methods.

Point 7: Section 2.2.1 is actually using the theory from Section 2.1.  How about Section 2.2.2?  How it relates with equation (1)?

Response 7:

The theory in Section 2.2.2 is to optimize  in Equation 1.

Point 8: In Section 2.2.1, it is better if the equation showing how the dehazed image is obtained from equation (2) is provided.

Response 8:

DCP provides an effective method to estimate model parameters. It can well estimate two important parameters in Equation 1: atmospheric illumination A and transmittance t(x). When A and t(x) are obtained, dehazing is completed by combining Equation 13 in Section 3.3.

Point 9: In equation (3), seems that coefficients a and b are locally determined, based on k.  How those values are calculated?  Better to give formulas.

Response 9:

To describe the derivation of guided filtering more clearly, add content to Page 4, Lines 142-147.

The description of this paragraph is mainly from reference [23]. Its function is to refine the initial transmittance, which can effectively remove the halo artifact, but will degrade the image quality. The shading partition enhancement mechanism proposed in this paper can effectively solve the problem of image quality degradation.

In Page 4, Lines 142-147.

The following sentences have been added

“To determine the linear coefficients   and , …”.

Point 10: What are Lw and Lb in equation (4)?

Response 10:

 and  are local bright and dark matrices, which store the enhanced bright and dark area information respectively.

In Page 6, Lines 216-218.

Based on computation, the brighter areas in the input image are enhanced and displayed in . As the enhanced dark areas are visible barely, this part will be inverted to show in .

Point 11: In equation (7), what will happen if bright equal to zero?

Response 11:

This process operates on bright area information. The area with bright=0 also needs to supplement the intermediate area information.

In Page 7, Equation 9.

 has been modified as .

Point 12: In equation (12), what is lg? Is this log_2 or log_10?

Response 12:

In Equation (12), lg denotes log10.

In Page 8, Equation 14.

 has been modified as .

Point 13: In line 246, it is claimed that the method is compared with "state-of-art methods".  Yet, [4] is from year 2016, [7] is from year 2021, [10] is from year 2015, [11] is from year 2018, and [12] is from year 2016.  From this, only ref. [7] is recent, whereas others are old methods.  Better to add more recent methods for comparisons (e.g., methods from year 2021 and 2022).

Response 13:

Datasets in the latest paper data are still unavailable so the STOS dataset was used as benchmark in this study. Insteadly, state-of-art methods will be modified as established dehazing methods.

Point 14: Methods in line 246 and 247 should be introduced properly in Section 1.  It would be nice if the authors could discuss some limitations of these methods in section 1.

Response 14:

The various methods used in the following texts and their limitations are introduced in Section 1, so that the reader can have a clearer understanding.

In Page 2, Lines 45-53.

The discussion has been added as follows: “DCP provides an effective method to estimate model parameters, …”. This part mainly introduces three kinds of image dehazing algorithms and the limitations of some of them.

Point 15: References are mostly old.  Better to add some more recent references, and discuss them in Section 1.  Some examples:

https://doi.org/10.3390/s20185170

https://doi.org/10.3390/s21237922

https://doi.org/10.3390/electronics10212636

Response 15:

References to newer papers can be a good way to increase the credibility of the article.

In Page 2, Lines 45-64.

The following sentences have been added

“Kim [6] proposes a sky detection method using region-based and boundary-based sky segmentation, which enables DCP to perform image restoration for sky of various shapes. ….”.

Reviewer 2 Report

Minor revision
This manuscript introduces a novel GIC image dehazing algorithm based on global illumination compensation mechanism. The GIC method compensates for the intensity of light scattered when light passes through atmospheric particles such as fog.  In summary, the research is interesting and provides valuable results, but the current document has several weaknesses that must be strengthened in order to obtain a documentary result that is equal to the value of the publication.

(1) Concerning the presentation of the contents, the document is acceptable. Nonetheless, it is recommended that authors develop proofreading to avoid common mistakes such as expression error. For example, is the word 'GIS' in line 140 correct ? If it is correct , please explain the abbreviation.
(2) The document contains a total of 23 employed references, of which 12 are publications produced in the last 5 years (52%), 8 in the last 5-10 years (35%), 3 than 10 years old (13%), implying a total percentage of 87 % recent references. In this way, the total number is insufficient, their actuality is high. 
(3) For the core ‘Global illumination compensation factor’ and ‘Dehazing process of the GIC algorithm’ of this paper, this paper has carried out A detailed description of the principle. Are they all innovative contents? If they are not, please point out the innovative part; if they are application innovation rather than principle innovation, please point out the prospect of innovative technology.
(4)The abstract is complete and well-structured and explains the contents of the document very well. Nonetheless, the part relating to the results could provide numerical indicators obtained in the research.
(5)In the part of the introduction of the research topic, the existing fog removal methods can be summarized into three categories, and the main contributions of this paper are also mentioned. All these are clear, and the novelty can be more highlighted if they can be compared. 
(6)Vision technology applications in various engineering fields, should also be introduced for a full glance of the scope of related area. The first paragraph introducing the research topic may present a much broad and comprehensive view of the problems related to your topic with citations to authority references (Binocular vision measurement and its application in full-field convex deformation of concrete-filled steel tubular columns; Seismic performance evaluation of recycled aggregate concrete-filled steel tubular columns with field strain detected via a novel mark-free vision method).
(7)In chapter 2, the atmospheric scattering model and the principle of the existing fog removal methods are introduced, and the shortcomings of the traditional methods are summed up. At the same time, the solutions are finally mentioned, which is very logical and clear.
(8)Chapter 3 introduces the concepts and principles of the shading partition enhancement global illumination compensation factor and evaluation indexes of image dehazing, but there is a lack of practical application methods and methods. If the methods to realize these two innovative parts in software resources can be added, it will be more convincing.
(9)In the experiment, the images processed by the shadow segmentation mechanism and the images processed by B1 and B2 methods are compared and analyzed, and evaluated with the indicators mentioned in Chapter 3. Please note whether B1 and B2 methods are the latest processing methods?
(10)It should mention the scope for further research as well as the implications/application of the study.
(11)I recommend including the limitations regarding the consideration of damage indicated in this review in the limitations assessment. This part of the document can be improved and completed with more rigour.

Author Response

Response to Reviewer 2 Comments

Point 1: Concerning the presentation of the contents, the document is acceptable. Nonetheless, it is recommended that authors develop proofreading to avoid common mistakes such as expression error. For example, is the word 'GIS' in line 140 correct ? If it is correct , please explain the abbreviation

Response 1: Similar typos have been corrected in the text.

Point 2: The document contains a total of 23 employed references, of which 12 are publications produced in the last 5 years (52%), 8 in the last 5-10 years (35%), 3 than 10 years old (13%), implying a total percentage of 87 % recent references. In this way, the total number is insufficient, their actuality is high.

Response 2: This paper contains a total of 30 employed references, of which 19 are publications produced in the last 5 years (63%), 8 in the last 5-10 years (27%), 3 than 10 years old (10%), implying a total percentage of 90% recent references.

Point 3: For the core ‘Global illumination compensation factor’ and ‘Dehazing process of the GIC algorithm’ of this paper, this paper has carried out A detailed description of the principle. Are they all innovative contents? If they are not, please point out the innovative part; if they are application innovation rather than principle innovation, please point out the prospect of innovative technology.

Response 3: In this study, the shading partition enhancement mechanism and the global illumination compensation factor can be regarded as application innovations. The former can reduce the influence of background on brightness and color before performing a priori analysis on the image. The latter can be combined with guided filtering to optimize the initial transmittance, resulting in a clearer outline of the transmittance map.

Point 4: The abstract is complete and well-structured and explains the contents of the document very well. Nonetheless, the part relating to the results could provide numerical indicators obtained in the research.

Response 4: The Abstract has been modified as follows:

“The GIC approach outperforms other dehazing methods in maintaining the naturalness of the image, and obtains improved PSNR and SSIM indexes.” has been modified as “Compared with the average metrics of each method, the peak signal-to-noise ratio (PSNR) and the structural similarity (SSIM) are improved by 3.25,0.084.”

Point 5: In the part of the introduction of the research topic, the existing fog removal methods can be summarized into three categories, and the main contributions of this paper are also mentioned. All these are clear, and the novelty can be more highlighted if they can be compared.

Response 5: In Page 2, Lines 45-64. The following sentences have been added

“DCP provides an effective method to estimate model parameters, …”. This part mainly introduces three kinds of image dehazing algorithms and the limitations of some of them.

Point 6: Vision technology applications in various engineering fields, should also be introduced for a full glance of the scope of related area. The first paragraph introducing the research topic may present a much broad and comprehensive view of the problems related to your topic with citations to authority references (Binocular vision measurement and its application in full-field convex deformation of concrete-filled steel tubular columns; Seismic performance evaluation of recycled aggregate concrete-filled steel tubular columns with field strain detected via a novel mark-free vision method).

Response 6: In Page 1, Lines 33-35.

“This results in the visual effect of a loss of contrast, ….” has been modified as “This results in a loss of contrast, visibility and vividness in images required for vision technology [1-2] due to the scattering effect of light through haze particles.”.

Point 7: In chapter 2, the atmospheric scattering model and the principle of the existing fog removal methods are introduced, and the shortcomings of the traditional methods are summed up. At the same time, the solutions are finally mentioned, which is very logical and clear.

Response 7: To obtain a comprehensive evaluation of the proposed GIC method, two candidate dehazing methods namely B3 and B4 have been selected to compare with the GIC method. More details can be found in the modified Table 1.

Point 8: Chapter 3 introduces the concepts and principles of the shading partition enhancement global illumination compensation factor and evaluation indexes of image dehazing, but there is a lack of practical application methods and methods. If the methods to realize these two innovative parts in software resources can be added, it will be more convincing.

Response 8:

Algorithm 1 The shading partition enhancement mechanism

1、Read image

2、Get the maximum and minimum values within an × area of the image.

From Equation (6)

3、Enhance the center pixel with surrounding pixels.

4、Get the range of bright and dark regions from the input image.

From Equation (7)(8)

5、Get enhanced bright and dark area information.

6、Image fusion

From Equation (9)

The function max_min() is employed to compute the maximum and minimum pixel values in the  ×  range. The function Img_sharp() represents the result of the multiplication of the surrounding pixels with the center pixel value. Functions of max() and min() are used to get the maximum and minimum values of a matrix. The mean() is used to get the mean of the matrix.

Algorithm 2 Global Illumination Compensation Factor

1、The DCP calculation is performed on the enhanced image.

From Lines 253

2、Calculates the average value of pixels in a small range.

Point 9: In the experiment, the images processed by the shadow segmentation mechanism and the images processed by B1 and B2 methods are compared and analyzed, and evaluated with the indicators mentioned in Chapter 3. Please note whether B1 and B2 methods are the latest processing methods?

Response 9: In Page 9, Lines 302-308.

The purpose of this part of the experiment is to highlight that the shading partition enhancement mechanism is more suitable for image dehazing than other image enhancements. Therefore, the GIC method is compared with the methods without image enhancement, histogram equalization, CLAHE, and Retinex. Four candidate dehazing methods namely B1, B2, B3 and B4 have been selected to compare with the proposed GIC method in Section 4.1. The following are qualitative and quantitative comparisons, which demonstrate the feasibility and effectiveness of the shading partition enhancement mechanism in the GIC method.

             (a)

      (b)

(c)

(d)

(e)

Figure 1. Comparison of outdoor image dehazing for the five methods. (a)B1 method processing results, (b) B2 method processing results, (c) B3 method processing results, (d) B4 method processing results, (e) GIC method processing results.

Original Table 1. Comparison of indexes of the three methods.

Methods

PSNR

SSIM

SSEQ

NIQE

AVG

SD

AVG

SD

AVG

SD

AVG

SD

B1

18.67

3.61

0.88

0.05

18.20

6.36

2.61

0.60

B2

18.48

3.25

0.79

0.09

18.01

6.30

3.39

0.81

GIC

22.07

3.02

0.90

0.05

15.49

5.60

2.76

0.59

New Table 1. Comparison of indexes of the five methods.

Methods

PSNR

SSIM

SSEQ

NIQE

AVG

SD

AVG

SD

AVG

SD

AVG

SD

B1

18.67

3.61

0.88

0.05

18.20

6.36

2.61

0.60

B2

18.48

3.25

0.79

0.09

18.01

6.30

3.39

0.81

B3

19.54

2.39

0.71

0.09

16.25

5.87

4.14

0.81

B4

13.75

4.10

0.51

0.24

24.65

15.10

4.09

1.72

GIC

22.07

3.02

0.90

0.05

15.49

5.60

2.76

0.59

Method B1 does not perform image enhancement, method B2 uses histogram equalization processing, method B3 denotes Contrast Limited Adaptive Histogram Equalization (CLAHE) processing, and method B4 adapts Retinex processing. Except for the different image enhancement methods, the other processes of these methods are the same as the GIC algorithm.

From both qualitative and quantitative perspectives, the shading partition enhancement mechanism has better dehazing effect and stability than other image enhancement methods, proving the effectiveness of this mechanism in image dehazing.

Point 10: It should mention the scope for further research as well as the implications/application of the study.

Response 10: This GIC method is proposed to deal with out-door images, especially those with blue sky or other consistent background. It can remove the fog effectively, thus improving the quality of images. In this case, downstream tasks including object detection and image classification can be accomplished with improved accuracy.

Point 11: I recommend including the limitations regarding the consideration of damage indicated in this review in the limitations assessment. This part of the document can be improved and completed with more rigour.

Response 11:

About the limitation assessment, we carefully compare established image dehazing and try to find potential improvement based on dehazing mechanism. In Paragraph 4 of Abstract, the description has been modified as follows:

Page 2, Line 65

Among the above methods, although the fusion-based method guarantees the gradient information of multi-scale input, its image restoration strength is insufficient to some degree. The deep learning based dehazing methods usually need considerable well-labeled samples, indicating high cost in data collection and curation. Based on the atmospheric scattering model, we analyze the shortcomings of the DCP method and the fog imaging principle. It is hypothesized that the brightness and color of a pixel of the input image will be affected by the surrounding pixels in a limited range. In order to eliminate negative interference from the neighborhood of pixel, this paper proposes the GIC algorithm to compute real prior information.

Reviewer 3 Report

This manuscript proposed a so called global illumination compensation approach to improve the dehazing quality upon traditional method in which the interference from the background is not considered. The proposed method exhibits better quantitative indices for some specific test pictures, such as PSNR, SSIM etc., compared to traditional methods, and can be considered as a technical progress for dehazing algorithm. Some comments are given as follows for the authors’ reference to improve the manuscript.

  1. The description is not quite clear, which makes the manuscript difficult to read. For example, line 132-136 describes the deficiency of traditional method, which is the important part to highlight the contribution of the proposed method. But the statement does not clearly describe what is exactly missing in the traditional method.
  2. The shading partition enhancement is the key contribution from the manuscript. The rest part of algorithm such as DCP already exists. How shading partition enhancement is related to resolving the issues from traditional method as described from 132-136 is not clearly described.
  3. The meaning of weighting factor in eq(10) is not clear, and how the value is determined in the test picture is not described either.
  4. As the proposed algorithm take the influence of background into consideration, it is reasonable to speculate that the final performance could be background dependent. It could perform better for some specific pictures, but worse for the others. Could a rigorous argument be provided to show that the performance will always be better than that of traditional methods.
  5. The “refracted” in line 123 and Fig. 1 should be “reflected” ?

Author Response

Response to Reviewer 3 Comments

Point 1: The description is not quite clear, which makes the manuscript difficult to read. For example, line 132-136 describes the deficiency of traditional method, which is the important part to highlight the contribution of the proposed method. But the statement does not clearly describe what is exactly missing in the traditional method.

Response 1:

When the conventional model-based methods directly perform prior analysis on the input images, certain problems will appear during image dehazing. These problems include but are not limited to:

(1) The values of A tend to be underestimated due to the reflection that occurs when light from the light source passes through atmospheric particles such as fog.

(2) t(x) refers to the part of the light reflected by the target pixel that is not scattered during the process of projecting the light from the target pixel to the camera through the medium. The influence of the surrounding pixels on the target pixel causes the part received by the camera to change, resulting in an inaccurate estimated t(x).

In Page 4-5, Lines 163-170, the manuscript has been modified as following:.

The atmospheric scattering model in Section 2.1 suggests that estimation of the atmospheric illumination and transmittance is two crucial steps to recover a haze-free image. In order to obtain more accurate atmospheric illumination value and transmittance, we propose GIC, an image dehazing algorithm based on atmospheric scattering model and considering image background interference. A few intermediate outputs in our procedure to obtain  from  using GIC method are shown in Figure 2.

Point 2: The shading partition enhancement is the key contribution from the manuscript. The rest part of algorithm such as DCP already exists. How shading partition enhancement is related to resolving the issues from traditional method as described from 132-136 is not clearly described.

Response 2:

By analyzing the pixels around the atmospheric illumination observation point obtained by DCP, the pixel value of the observation point tends to increase suddenly compared with the surrounding pixel value. Therefore, we believe that this phenomenon is due to the fact that the pixel value of the observation point is affected by the surrounding pixels.

In Page 4, Lines 185-189, the manuscript has been modified as following:

“We found that the brightness standard deviation of surrounding pixels tends to fluctuate within a small range. After adding the brightness value of the center pixel, the standard deviation increases significantly. Combined with the experimental results, we believe that the sudden increase in the brightness value of the center pixel is affected by the surrounding pixels.”

In Page 9, Lines 274-282, the manuscript has been modified as following:

The shading partition enhancement mechanism is essentially an image enhancement algorithm, which can effectively compensate for the light intensity reflected by the light of the light source when it passes through atmospheric particles such as fog. It can be seen from Figure 3 and Eq (6) that the calculation method of this mechanism simulates the influence of surrounding pixels on the target pixel, so that the image is restored to the state without background interference, so as to prepare for the subsequent DCP calculation processing.

The experimental results show that the surrounding pixel values of the atmospheric observation points obtained by analyzing the GIC algorithm are close to or even equal to the pixel values of the observation point. In a certain area centered on this point, its pixel value is the maximum value. Therefore, the observation point of atmospheric illumination can be found more accurately after processing by this mechanism.

Point 3: The meaning of weighting factor in eq(10) is not clear, and how the value is determined in the test picture is not described either.

Response 3:

In Page 7, Lines 253-255. In-depth descriptions about Eq(10) have been added.

The global illumination compensation factor is used after the optimization of the initial transmittance through guided filtering, so that the transmittance image can have a clearer outline. It is calculated by taking the average value of pixels in a small range centered on  as the weight of  in the dark channel of the enhanced image .

Point 4: As the proposed algorithm take the influence of background into consideration, it is reasonable to speculate that the final performance could be background dependent. It could perform better for some specific pictures, but worse for the others. Could a rigorous argument be provided to show that the performance will always be better than that of traditional methods.

Response 4:

For synthetic indoor images, we process and analyze the foggy images with various fog concentration ranging from 1 to 10. The GIC algorithm has an satisfactory dehazing effect at low concentration fog, i.e. 1 to 3, but the effect will gradually decrease as the fog concentration increases. For outdoor real images, the GIC algorithm has an improved dehazing personsefor the SOTS data set alone, due to the lack of image data of different concentration levels.

In response to the questions raised by the reviewers, we analyzed the indicator data. Only a few images outperform the candidate methods on all four metrics. So we are temporarily missing a rigorous argument to show that the performance will always outperform traditional methods. This will be an optimization direction for us in the future.

The shading partition enhancement mechanism is a critical mechanism of the GIC dehazing algorithm. In order to better highlight its role in image dehazing, we added two image enhancement algorithms i.e. B3 and B4 that were proposed in recent years, to illustrate the pros-and-cons of these dehazing approaches. the method B3 denotes Contrast Limited Adaptive Histogram Equalization (CLAHE) processing, and method B4 adapts the Retinex processing. The following are qualitative and quantitative comparisons, both of which demonstrate the feasibility of the shading partition enhancement mechanism in image dehazing.

In Page 9, Lines 302-308, additional experiments have been accomplished to illustrate the characteristics of the proposed GIC method. Relevant experimental analysis are given as following:

(a)

(b)                  (c)

(d)

(e)

Figure 1. Comparison of outdoor image dehazing for the five methods. (a)B1 method processing results, (b) B2 method processing results, (c) B3 method processing results, (d) B4 method processing results, (e) GIC method processing results.

Original Table 1. Comparison of indexes of the three methods.

Methods

PSNR

SSIM

SSEQ

NIQE

AVG

SD

AVG

SD

AVG

SD

AVG

SD

B1

18.67

3.61

0.88

0.05

18.20

6.36

2.61

0.60

B2

18.48

3.25

0.79

0.09

18.01

6.30

3.39

0.81

GIC

22.07

3.02

0.90

0.05

15.49

5.60

2.76

0.59

has been modified to

Table 1. Comparison of indexes of the five methods.

Methods

PSNR

SSIM

SSEQ

NIQE

AVG

SD

AVG

SD

AVG

SD

AVG

SD

B1

18.67

3.61

0.88

0.05

18.20

6.36

2.61

0.60

B2

18.48

3.25

0.79

0.09

18.01

6.30

3.39

0.81

B3

19.54

2.39

0.71

0.09

16.25

5.87

4.14

0.81

B4

13.75

4.10

0.51

0.24

24.65

15.10

4.09

1.72

GIC

22.07

3.02

0.90

0.05

15.49

5.60

2.76

0.59

Method B1 does not perform image enhancement, method B2 uses histogram equalization processing, method B3 denotes Contrast Limited Adaptive Histogram Equalization (CLAHE) processing, and method B4 adapts Retinex processing. Except for the different image enhancement methods, the other processes of these methods are the same as the GIC algorithm.

From both qualitative and quantitative perspectives, the shading partition enhancement mechanism has better dehazing effect and stability than other image enhancement methods, proving the effectiveness of this mechanism in image dehazing.

Point 5: The “refracted” in line 123 and Fig. 1 should be “reflected” ?

Response 5:

In Page 4, Lines 156.

“refracted” has been modified as “reflected”.

Other grammatical errors were also modified as well.

Round 2

Reviewer 1 Report

I am satisfied with the improvements done by the authors.

Reviewer 3 Report

The authors have addressed the issues raised in the previous review, and the manuscript is recommended for the publication in Sensors